# Visiting Distant Neighbors in Graph Convolutional Networks

## Abstract

We extend the graph convolutional network method for deep learning on graph data to higher order in terms of neighboring nodes. In order to construct representations for a node in a graph, in addition to the features of the node and its immediate neighboring nodes, we also include more distant nodes in the calculations. In experimenting with a number of publicly available citation graph datasets, we show that this higher order neighbor visiting pays off by outperforming the original model especially when we have a limited number of available labeled data points for the training of the model.

## 1 Introduction

The problem of learning representations in a graph has been a major attraction in representation learning for a while now. Several algorithms have been developed trying to represent the information from node features, edge features, and adjacency matrix of a graph with a low dimensional vector in order to enable further analysis on them (Grover & Leskovec, 2016; Zhu et al., 2003; Belkin et al., 2006; Ahmed et al., 2022; Defferrard et al., 2016).

A semi-supervised learning method on graph-structured problem is defined when we are looking to find the missing class labels for nodes (e.g. published papers) in a graph of relations (e.g. citation network of those published papers) based on a small subset of known labels for nodes. These types of problems can also be found when one is trying to cluster nodes of a graph into similar groups. Roughly speaking, by using these learning techniques, we can obtain lower dimensional embedding vectors for nodes in a graph where we can apply simple geometry distance functions to quantify the similarity or dissimilarity between two nodes.

Graph convolutional networks were introduced by Kipf & Welling (2016) as a method for aggregating the features of the node and its neighbors in a convolving neural network layer in order to obtain embedding vectors. The main formula behind a first-neighbor graph convolutional network is encapsulated in the following formula:

$$H^{(l+1)} = \sigma(\tilde{D}^{-\frac{1}{2}} \tilde{A} \tilde{D}^{-\frac{1}{2}} H^l W^l) \tag{1}$$

Where $\tilde{A} = A + I_N$ is the adjacency matrix of the original undirected graph $\mathcal{G}$ (for which we are trying to learn the node embeddings) with the self-loops added in order to capture the feature vector of the node. $\tilde{D} = D + I_N$ is the degree matrix of the graph, where the diagonal element $D_{ii}$ is the degree of the node $i$, plus one to account for the self-loop. $H^l$ is the feature matrix in the $l$th layer and $H^0 = X$ is simply the feature vectors of each node. $W^l$ is the trainable weight matrix in the $l$th layer where its dimensions denote the feature vector and the output dimensions of the convolutional layer and finally $\sigma$ stands for a non-linear activation function such as the rectified linear unit (ReLU) to introduce non-linearity into the model. This model, alongside with many other representation learning methods for graphs, assumes that the connections in a graph are signs of node similarity, which is not always the case in graph-structured datasets.

In previous studies done on real-world graphs, it is demonstrated how important it is to take into account the global role of a node in a graph when trying to make predictions about it. As our contribution, in this work we will expand the notion of graph convolutional networks to higher order neighbors of nodes in the graph in order to capture long-range correlations between nodes and let the optimization method decide for

the best coefficients for different sets of neighbors and node's own feature vectors. Then we report the results of this new method on a number of datasets to demonstrate how this higher order approximation affects the performance of the model in terms of accuracy and efficiency, particularly in generalizing the model from a lower number of labeled data. We will also experiment on random graphs in order to measure wall-clock time performance against the original model.

## 2 Higher Order Neighbor Convolutions on Graphs

In this section we introduce our graph neural network model which is an expansion of the previously known GCN to the $n$th neighborhood:

$$H^1 = \sigma(\sum_{i=0}^{n} \gamma_i \bar{A}_i H^0 W) \tag{2}$$

Where $\gamma_i$ is a trainable coefficients for the $i$th neighborhood of the node, $W$ is the trainable weight matrix, $H^0 = X$ are the node features and $H^1$ is the resulting node representations from this layer. $\bar{A}_i$ is the normalized adjacency matrix of $i$th neighborhood where the sum of each row is equal to 1 to avoid the dominance of high-degree nodes in the graph and $(\bar{A}_i)_{ab} = 1$ if the shortest path between two nodes is equal to $i$, excluding the self-loops and $(\bar{A}_i)_{ab} = 0$ otherwise. In this definition, $\bar{A}_0$ would be the identity matrix, $\bar{A}_1$ would be the normalized adjacency matrix and $\bar{A}_2$ would have elements equal to 1 where the shortest path between two nodes is equal to 2 and so on. Note that this can be considered similar to expanding the kernel size in a image processing convolutional neural network from 3 to 4 and larger. Computing $\bar{A}_i$s would be computationally cheaper than finding the matrix of shortest paths, since we will be stopping the approximation on a certain distance (e.g. 2 for second order neighborhood approximation).

The expanded propagation rule up to the second neighborhood approximation would look like the following:

$$H^1 = \sigma((\gamma_0 I + \gamma_1 \bar{A}_1 + \gamma_2 \bar{A}_2) H^0 W) \tag{3}$$

## 3 An Experiment in Semi-Supervised Node Classification

Now that we have a propagation rule for this expanded model, in order to compare the results with the first-neighborhood model, we turn to the problem of semi-supervised classification of nodes in a graph, where we only have known labels for a tiny portion of the nodes and we want to classify other nodes. We will be using graph-structured datasets from citation networks of different sizes. The objective is to classify graph nodes into different classes by learning from only a few labeled nodes. The claim is that the information from node features, combined with the information from the adjacency matrix of the graph can result in a better representation learning for this task.

### 3.1 Model Architectures

Here we define two graph neural networks with two layers of our expanded GCN layers on an undirected graph $\mathcal{G}$ up to the second and third neighbor approximation. The forward model for these neural networks would take the following forms:

$$Z_2 = \text{softmax}((\gamma_0^1 I + \gamma_1^1 \bar{A}_1 + \gamma_2^1 \bar{A}_2) \, \text{ReLU}(\gamma_0^0 I + \gamma_1^0 \bar{A}_1 + \gamma_2^0 \bar{A}_2) \, X \, W^0) \, W^1) \tag{4}$$

$$Z_3 = \text{softmax}((\gamma_0^1 I + \gamma_1^1 \bar{A}_1 + \gamma_2^1 \bar{A}_2 + \gamma_3^1 \bar{A}_3) \, \text{ReLU}(\gamma_0^0 I + \gamma_1^0 \bar{A}_1 + \gamma_2^0 \bar{A}_2 + \gamma_3^0 \bar{A}_3) \, X \, W^0) \, W^1) \tag{5}$$

Where $W^0 \in \mathbb{R}^{F \times L}$ and $W^1 \in \mathbb{R}^{L \times O}$ with $F$ being the number of features of the nodes, $L$ being the dimension of the hidden layer, and $O$ the dimension of the embedding vectors. The weights in the model are then

optimized by gradient descent using Adam stochastic optimization and we include a dropout (Srivastava et al., 2014) in the network to improve generalization. We will use the negative log loss likelihood as the loss function here. The pre-processing step would include calculating the $\bar{A}_1$, $\bar{A}_2$, and $\bar{A}_3$ matrices and normalizing the feature vectors.

### 3.2 Related Works and Baseline Model

Several previous approaches for the problem of learning graph representations have been studied in the past. Some of the classical methods such as the label propagation and manifold regularization on graphs, take advantage of graph Laplacian regularization and have been inspired by rigorous graph theory. But more recent methods, inspired by the success of deep learning models in different fields, can learn node representations by sampling from different types of random-walks across the graph such as DeepWalk, node2vec, role2vec, and Planetoid.

Although the scheme of neural networks on graph was studied before (Gori et al., 2005; Scarselli et al., 2009), graph convolutional neural networks in their current form were introduced by Duvenaud et al. (2015) and further studied in works by Niepert et al. (2016); Bruna et al. (2014), introducing a spectral graph convolutional neural network and introduction of a fast local convolution by Hammond et al. (2009).

In a closely related work by Atwood & Towsley (2015), the authors use diffusion across graph nodes in order to calculate the convolutional representations for nodes. Our work is different in the sense that we also allow the model to optimize the coefficients of the contributions of different neighborhood layers and the features of the node itself based on the problem. This would be more significant when dealing with graphs where, for example, the features of the 2nd-hop neighbors are also important. As a simple example, triangles have been prove to play an important role in community structures of experimental networks (Klymko et al., 2014; Oloomi et al., 2021). Examples of these graphs would be knowledge graphs such as transaction networks where we are looking for specific types of frauds such as money-laundering. In these networks the multi-hop structure loops are important to identify fraudulent motifs (Singh & Best, 2019). A similar idea has also been studied in Abu-El-Haija et al. (2019) where the authors assign different weight matrices for different neighborhood distances, whereas in this work we will be using the same weight matrix for different neighborhoods but as a linear combination. Also notice how our proposed expansion is different from just stacking up more vanilla GCN layers in order to get distant neighbor aggregation (like the study by Li et al. (2022)) in terms of the number of parameters and model complexity since we are using the same weight matrix for different neighborhood distances and just add a linear combination coefficient.

Distant neighborhood aggregation is not just limited to graph convolutional networks, but also studied in random walks and other methods as in Xu et al. (2018). In this work, the authors develop a local-structure-aware framework for involving the information from farther nodes in different graph representation learning models. Also the authors in Yang et al. (2022) modify the graph neural networks to remove the noise from irrelevant propagations by taking into account the structural role of a node using eigenvalue decomposition of the graph. Which is similar to our work in terms of extending the graph embeddings to higher order relations.

Since based on previous experiments on their paper (Kipf & Welling, 2016), the original graph convolutional networks (which considers only the first neighbors in convolution) outperforms the classical models and early deep models (DeepWalk, node2vec, Planetoid) on the same task as here, in order to demonstrate the performance gain acquired from considering farther neighbors, we will only compare our results to the original GCN. We will be using the similar neural network structure used in the original paper, which is a two layer GCN with ReLU activation function.

### 3.3 Experimental Setup

We will be testing the performance of the models on publicly available citation network datasets, alongside with artificial random datasets for wall-clock time performance analysis.

Table 1: Dataset statistics

| Dataset | Nodes | Edges | Classes | Features |
|---------|-------|-------|---------|----------|
| CITESEER | 3327 | 4732 | 6 | 3703 |
| CORA | 2708 | 5409 | 7 | 1433 |
| PUBMED | 19717 | 44338 | 3 | 500 |

Table 2: Training hyperparameters

| Dropout | L2 | Output Dimension | Learning Rate |
|---------|-----|------------------|---------------|
| 0.5 | 5E-4 | 16 | 0.01 |

### 3.3.1  Datasets

Datasets used for model performance evaluation are presented in Table 1. Citeseer, Cora, and Pubmed are citation network datasets where each node represents published papers and a directed link from one node to another translates to a citation of the second paper in the first one. In these datasets, node features are bag-of-words vectors for each paper.

Note that in all the datasets we are neglecting the direction of edges and consider an undirected graph for the learning task.

### 3.3.2  Training and Testing Procedure

In order to compare the ability of each model in generalizing from the training data, we will treat the number of available labelled nodes as a parameter. So we will be training each model on different number of available nodes per class and then measure their performance on the a balanced set of the remaining nodes. We will be using 1, 2, 5, 10, 15, and 20 nodes per class for training, 30 nodes for validation and stopping the training, and the rest of the nodes (in a balanced way between classes) for measuring the accuracy. Note that in each repetition of the experiment, all of these nodes are shuffled randomly. We continue the training for a maximum number 200 epochs and use early stopping on 20 epochs. Meaning that if the validation accuracy does not improve after any 20 consecutive epochs, the training will be halted. Other training hyperparameters are presented in Table 2. We repeat training and testing for each model for a total number of 500 different random initializations from Glorot & Bengio (2010) and report the mean and the standard deviation of the accuracy as results.

### 3.3.3  Implementation

We will be using PyTorch Paszke et al. (2019) for implementing the models to work on a GPU accelerated fashion and we will make use of sparse matrix multiplications in PyTorch, which results in a complexity of $\mathcal{O}(\mathcal{E})$ i.e. linear in number of non-zero matrix elements.

## 4  Results

Results for different datasets and number of available labeled nodes are presented in Table 3. GCN is the original graph convolutional network, GCN-2 represents the network with layers expanding up to the second neighborhood in Equation 4, and GCN-3 is expanding up to the third neighborhood in Equation 5. The results are in agreement with the validity of the expansion along neighborhood size, meaning that the accuracy of the model remains the same or increases with the graph convolution neighborhood size. When a lower number of training datapoints are available, higher order models outperform the original GCN by a larger margin but in abundance of training datapoints, we observe a saturated similar accuracy amongst the different models.

Table 3: Accuracy and the standard error of the models on different datasets with various number of available nodes for training. A higher order model always outperforms or matches the accuracy of a lower order model.

| $n = 1$ | Cora | Citeseer | Pubmed |
|---|---|---|---|
| GCN | $35.5_{\pm 0.4}$ | $24.5_{\pm 0.3}$ | $50.2_{\pm 0.5}$ |
| GCN-2 | $44.5_{\pm 0.4}$ | $28.3_{\pm 0.3}$ | $53.8_{\pm 0.5}$ |
| GCN-3 | $48.2_{\pm 0.4}$ | $30.8_{\pm 0.3}$ | $54.7_{\pm 0.7}$ |
| $n = 2$ | | | |
| GCN | $49.9_{\pm 0.3}$ | $31.8_{\pm 0.4}$ | $59.3_{\pm 0.4}$ |
| GCN-2 | $57.8_{\pm 0.3}$ | $36.2_{\pm 0.3}$ | $61.3_{\pm 0.4}$ |
| GCN-3 | $61.5_{\pm 0.3}$ | $40.1_{\pm 0.3}$ | $60.6_{\pm 0.6}$ |
| $n = 5$ | | | |
| GCN | $68.0_{\pm 0.3}$ | $46.9_{\pm 0.4}$ | $68.3_{\pm 0.3}$ |
| GCN-2 | $71.1_{\pm 0.2}$ | $50.0_{\pm 0.3}$ | $68.8_{\pm 0.2}$ |
| GCN-3 | $71.5_{\pm 0.2}$ | $51.4_{\pm 0.2}$ | $69.4_{\pm 0.3}$ |
| $n = 10$ | | | |
| GCN | $75.9_{\pm 0.2}$ | $57.4_{\pm 0.3}$ | $72.8_{\pm 0.2}$ |
| GCN-2 | $76.6_{\pm 0.1}$ | $57.8_{\pm 0.2}$ | $73.2_{\pm 0.1}$ |
| GCN-3 | $76.6_{\pm 0.1}$ | $57.9_{\pm 0.1}$ | $73.5_{\pm 0.2}$ |
| $n = 15$ | | | |
| GCN | $78.3_{\pm 0.2}$ | $61.3_{\pm 0.2}$ | $74.6_{\pm 0.1}$ |
| GCN-2 | $78.7_{\pm 0.1}$ | $61.4_{\pm 0.1}$ | $75.0_{\pm 0.2}$ |
| GCN-3 | $78.9_{\pm 0.1}$ | $61.2_{\pm 0.1}$ | $74.8_{\pm 0.2}$ |
| $n = 20$ | | | |
| GCN | $79.8_{\pm 0.2}$ | $63.7_{\pm 0.2}$ | $75.5_{\pm 0.1}$ |
| GCN-2 | $80.0_{\pm 0.1}$ | $63.6_{\pm 0.1}$ | $76.3_{\pm 0.1}$ |
| GCN-3 | $80.1_{\pm 0.1}$ | $63.4_{\pm 0.1}$ | $76.1_{\pm 0.1}$ |

The average training time per epoch on random graphs of size $N$ with $2N$ edges for different models are presented in Figure 4. Since the preprocessing step of calculating $\bar{A}_i$ matrices is only done once on each dataset, we are omitting this preprocessing step from the training time analysis.

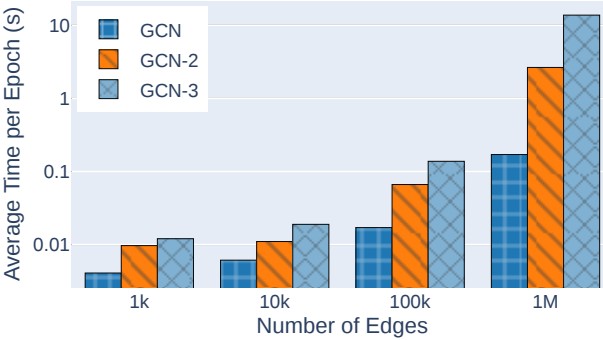

Figure 1: Wall-clock training time per epoch for different order of neighborhood and different random graphs of $N$ nodes and $2N$ edges.

Looking at the performance gain per epoch, we can see that some of this higher computation cost is compensated by a faster learning in the expanded GCN models. The data in Figure 4 is acquired by

averaging the accuracy per epoch for 100 different random initializations of the models on the Cora dataset. This shows that the expanded models would reach the same performance in a significantly lower number of epochs.

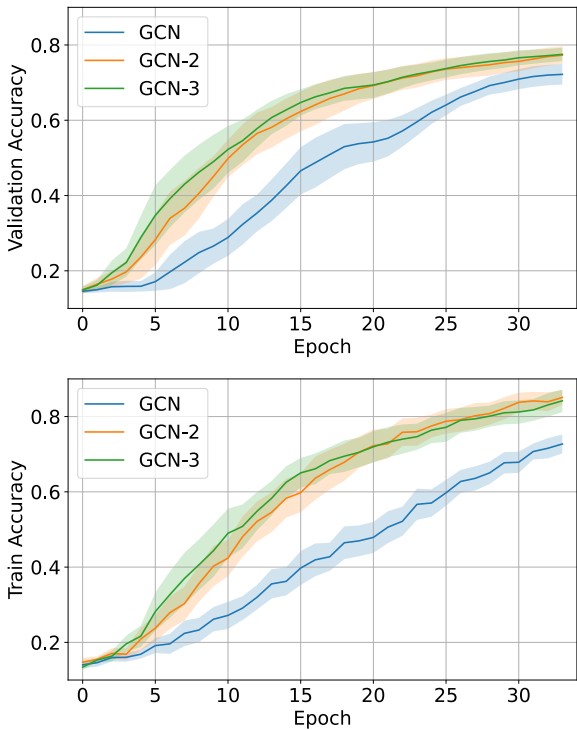

Figure 2: Mean accuracy of each model per epoch when training on the Cora dataset. The curves show average and standard deviation of accuracy on each training epoch on validation and training data. We can see how the expanded models tend to learn faster from the data.

## 5 Future Work and Conclusion

The introduced expanded model has several limitations which may be improved in the future works. The current model does not include edge features and edge weights natively. There are several possible workarounds for this issue, such as converting the edges into nodes with features but including the edge features in the convolution, or similar solutions to the work done by Hu et al. (2019). Expansion of the current gradient descent optimization to a mini-batch stochastic one, such as Gasteiger et al. (2022) would also be helpful for larger datasets where memory limitations would not allow full-batch calculations.

In this work we have expanded the current notion of graph convolutional neural networks to a model which considers different coefficients for node's self-features and each layer of neighborhood. This would help remove two original assumptions in this model where only the features of the node itself and the first neighbors (with similar coefficients) where used to learn node representations. Our model's propagation rule is expandable in terms of neighborhood distance and experiments on several datasets show a better generalization capability of this model compared to the original GCN, without adding to the trainable parameters, and particularly with a low number of available training nodes.

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
