# OpenReview forum: "Visiting Distant Neighbors in Graph Convolutional Networks"
_TMLR — Rejected by TMLR_

### Review · Reviewer_P3uD · 2023-04-25

**Summary Of Contributions:**

Claimed contribution: "We extend the graph convolutional network method for deep learning on graph data to higher order in terms of neighboring nodes."



**Audience:**

No

**Broader Impact Concerns:**

No ethical concerns.

**Claims And Evidence:**

No

**Requested Changes:**

Unfortunately, I don't think this work can be salvaged. There is no novelty, it only seems to be a learning experience for the authors.

**Strengths And Weaknesses:**

Unfortunately I can't recommend of accepting this paper. The claimed contribution itself has been studied for a few years and it is quite a mature subject. I don't understand how the authors claim that "they extend the graph convolutional network method for deep learning on graph data to higher order in terms of neighboring nodes". There are numerous papers that have done this already. For example:

[1] https://arxiv.org/abs/1907.06051

[2] https://arxiv.org/abs/1905.00067 (also cited by the authors)

[3] https://arxiv.org/abs/2009.14332

[4] https://arxiv.org/abs/1905.11136

[5] https://arxiv.org/abs/2205.13328

The above is just a small list, you can find more papers in the references of the above list of papers.

The authors only attempt to distinguish their work compared to [2], but the only distinction seems be that the authors use the same parameters for all convolutions. Also, in [2] one can also use the same parameters. So in view of this, the only novelty seems to be an experiment of using same parameters for all convolutions that is missing from [2]. Given this and that the paper's literature review lacks significantly, personally, I can't accept this as novel research

---

> ### Author Response · Authors · 2023-05-18
>
> Thank you for your time reading the manuscript and your review.
> Your first suggested work is different since it uses a multi layer perceptron, the second one is mentioned on our manuscript. The third reference extends the graph attention networks beyond the first neighborhood, unlike our work which is on graph convolutional networks. The 4th referred work is more widely studied in [1] and it is mostly on the similarities between the Weisfeiler-Lehman isomorphism test and GNNs. I do not see how it may be overlapping with extending GCNs beyond first neighborhood in a single layer. The last work is most related and explores the theoretical basis of K-hop aggregation. I believe it is a rigorous theoretical approach to the experimental study done on our work. It should be added to our manuscript and the fact that their arxiv publication date was so close to ours made us miss it. But the differences in terms of learnable layer coefficient and using the same weight matrix across neighborhoods persists between our work and this.
>
> [1]: https://arxiv.org/pdf/1810.00826.pdf

---

### Review · Reviewer_dnkc · 2023-04-28

**Summary Of Contributions:**

GCN is modeled as $H^1 = \sigma(\hat{A}H^0W)$ where $\hat{A}$ is the sym-normalized graph, $H^0 = X$ is the node feature matrix and $W$ is the learnable layer weight. In the paper, the authors propose to extend this idea as follows: $H^1 = \sigma(\sum_{i=1}^n \hat{A}_iH^0W)$ where $A_i$ captures the node information from the $i^{\textrm{th}}$ hop.

**Audience:**

Yes

**Claims And Evidence:**

No

**Requested Changes:**

Based on the above review, I would request the following changes to the paper. Note that all the points made below are critical.

1. **Clarify the relationship with Chebyshev polynomials:** Explicitly discuss the similarities and differences between the proposed approach and Chebyshev polynomials. One may follow the example in [A].

2. **Expand comparisons:** Include comparisons with Chebyshev polynomials, cited related works and other relevant methods from the filter learning literature [C, D, E]. This will help readers better understand the advantages of the proposed approach when considering incorporating distant neighbor information.

3. **Explain performance gaps:** Provide a detailed explanation for why the proposed approach performs well, despite the gap with Chebyshev polynomials, and how it differs from the improvements made in [B]. It would be good to add any ablation study that can corroborate the explanations.

4. **Provide insights into fast convergence:** Investigate and explain the reasons behind the faster convergence of the proposed model compared to other methods. It would be good to add any ablation study that can corroborate the explanations.

5. **Elaborate on low label rate performance:** Offer a clear explanation for why higher-order information is more helpful than nearby node information in low label rate settings, particularly the n=1 setting. Again, it would be good to add any ablation study that can corroborate the explanations.

6. **Incorporate missing related work:** Include the missing related work [C, D, E] in the related work section and discuss how these methods relate to the proposed approach.

By addressing these suggestions, the author can strengthen their paper by providing a clearer understanding of the proposed method's relationship to existing work, offering explanations for its performance, and showcasing its advantages through more comprehensive comparisons.

References:
* [A] Semi-Supervised Classification with Graph Convolutional Networks, ICLR 2017
* [B] Convolutional Neural Networks on Graphs with Chebyshev Approximation, Revisited, NeurIPS 2022
* [C] Adaptive Universal Generalized PageRank Graph Neural Network, ICLR 2021
* [D] BernNet: Learning Arbitrary Graph Spectral Filters via Bernstein Approximation, NeurIPS 2021
* [E] A Piece-wise Polynomial Filtering Approach for Graph Neural Networks, ECML-PKDD 2022

**Strengths And Weaknesses:**

Strengths:
1. **Simplicity:** The proposed approach is elegantly simple.
2. **Strong Performance in Low Label Rate Setting:** The results appear promising for the extremely low label rate setting.
3. **Fast Convergence in High Label Rate Setting:** Even for high label rate settings, where the performance of the proposed model becomes comparable with GCN, the proposed approach seems to offer faster convergence.

Weaknesses:
1. **Similarity to Pre-existing Idea:** The proposed idea seems to exist in a different form. In the GCN [A] paper, the authors began with a Chebyshev polynomial filter and derived the GCN model as the first-order approximation of the polynomial. Consequently, the proposed model could be viewed as a higher-order approximation of the Chebyshev polynomial with learnable coefficients.
2. **Lack of Comparison with Chebyshev Polynomials:** Given the similarity to the Chebyshev polynomials, it would be appropriate for the authors to compare their model not only with GCN but also with Chebyshev polynomials. It is known that the Chebyshev filter does not perform as well as GCN [A]. More recent papers have addressed this issue by introducing learnable coefficients [B].
3. **Insufficient Explanation for Success Despite Similarity with Chebyshev:** Significant effort was made in [B] to improve the performance of Chebyshev polynomials. Therefore, it is surprising that the proposed approach works well without a clear connection to these improvements.
4. **Unexplained Fast Convergence:** The proposed model converges faster, but the authors do not provide an explanation for this phenomenon.
5. **Unexplained Low Label Rate Performance:** The proposed approach demonstrates strong performance in low label rate settings, especially the n=1 setting. Given that there is only one label available per class, it is unclear why higher-order information would offer any benefit over nearby node information.
6. **Omission of Relevant Literature:** Several ideas have been explored in filter learning literature for incorporating information from distant neighborhoods and have been tested on heterophilic datasets [C, D, E]. These studies are not included in the related work section.
7. **Missing Comparative Analysis:** The paper cites various other works in the related work section but does not compare the proposed approach against any of them to gauge its relative advantages. This also applies to the omitted related work discussed above.

References:
* [A] Semi-Supervised Classification with Graph Convolutional Networks, ICLR 2017
* [B] Convolutional Neural Networks on Graphs with Chebyshev Approximation, Revisited, NeurIPS 2022
* [C] Adaptive Universal Generalized PageRank Graph Neural Network, ICLR 2021
* [D] BernNet: Learning Arbitrary Graph Spectral Filters via Bernstein Approximation, NeurIPS 2021
* [E] A Piece-wise Polynomial Filtering Approach for Graph Neural Networks, ECML-PKDD 2022

---

> ### Author Response · Authors · 2023-05-18
>
> Thank you for taking the time and reviewing the manuscript.
> The requested changes are helpful and they can be included in the manuscript. We will clarify the differences between this method and Chebyshev polynomials and add performance comparisons. We can also include backing up studies for the observed performance gaps and faster convergence. The missing related work will also be present in the next version of the manuscript.

---

### Review · Reviewer_hvcD · 2023-05-18

**Summary Of Contributions:**

The authors propose a method that extends 1-hop graph convolutional neural networks into a multiple hop GCN.

The aggregation rule is based on first visiting the k hop neighbors, learning a filter for each of them and also a scaling coefficient.

The authors demonstrate their method on the CORA/CITESEER/PUBMED datasets, showing some improvement.

**Audience:**

Yes

**Broader Impact Concerns:**

I find no broader impact concerns.

**Claims And Evidence:**

No

**Requested Changes:**

Please change your paper according to my review and questions above.

**Strengths And Weaknesses:**

Strengths:

- The authors explain the motivation of why such a method is needed
- The method is clear

Weaknesses:

- A more thorough experimental evaluation needs to be made. It is hard to draw conclusions from 3 small datasets.
The paper presents relatively low accuracy on baseline models such as GCN, which is not in line with the results showing in the GCN paper.
- More methods need to be compared with to show the effectiveness of the method.
Some missing related work:
1. In ChebNet (Defferrard et al) it was also proposed to consider multi hop graph kernels. What is the main difference between your work and ChebNet?
2. Similar to 1, which is well know, there are other more recent papers that propose multi hop kernels, such as [1] and [2]. Can you please discuss the differences?

- At the end of page 1, the authors state "In previous studies done on real-world graphs, it is demonstrated how important it is to take into account the global role of a node in a graph when trying to make predictions about it." Can the authors please add references?

[1] Path Integral Based Convolution and Pooling for Graph Neural Networks

[2]pathGCN: Learning General Graph Spatial Operators from Paths

---

> ### Author Response · Authors · 2023-05-18
>
> Thank you for your time reviewing the manuscript.
>
> To address your concerns and issues:
> 1- The results from the original GCN paper are different due to the sampling method we have chosen. The original GCN paper does not shuffle the train/test nodes, they used a standard fixed set for training each time but we did and we averaged the results.
>
> 2- The results from other common graph classification methods can be added, they were omitted to keep the paper more simple since the performances were not so comparable.
> ChebyNet is the more general framework where the idea of expansion beyond first neighborhood is studied. The main difference is the motivation and simplicity. ChebyNet is based on applying signal processing methods in order to define a localized filter similar to convolutional filters in CNNs whereas our work is taking a different approach to the same problem, but by expanding the linear model (GCN) without adding more hyperparametrs. We used the assumption that higher-order neighborhood is not different from instant neighborhood in terms of coefficients and we use the same weight matrix for different k-layers. The other difference is that we have added learnable coefficients for different neighborhoods which was not included in the ChebyNet.
>
> 3- Thank your for observing this issue. Examples and references have been provided in section 3.2 but we should back this sentence with references as well. The references will be added to the mentioned sentence.

---

### Review · Reviewer_VPzj · 2023-05-26

**Summary Of Contributions:**

The paper deals with the problem of utilizing higher-order information in the message passing mechanism of GNNs. Specifically, it introduces a variant of the GCN model, where different k-hop instances of the adjacency matrix are used in the propagation rule. The performance of the proposed methodology is empirically evaluated on three benchmark node classification datasets and is compared against the GCN model.

**Audience:**

Yes

**Broader Impact Concerns:**

-

**Claims And Evidence:**

No

**Requested Changes:**

In my comments above, I have included several changes that need to be made in the paper.

However, considering the limitations of the paper regarding the novel aspects of the methodology and the fact that similar ideas have been proposed in the literature, I am not confident that the paper could meet the expectations of TMLR.

**Strengths And Weaknesses:**

**Strengths**
- Simple model, well-motivated.
- Good performance results.

**Weaknesses**
- There are several issues with the paper. Although the results are interesting, the proposed methodology lacks novelty. Essentially, the propagation rule here relies on the powers of the adjacency matrix. The problem of incorporating higher-order information in GNNs has been widely studied in the past (e.g., [1], [2], [3]), and different formulations have been proposed.
- The paper misses theoretical support and explanations about the proposed propagation rule. Some of its properties related to convergence have been empirically studied. However, considering the fact that several papers follow a similar formulation, it would be interesting to theoretically study the properties of the model, including its expressive power and scalability. There is also a close connection to spectral GNN models, where such problems have been studied in the past ([4]).
- The empirical analysis part of the paper should be enhanced. Many different models improve the performance of GCN, thus, more baseline models should be used. Also, adding an ablation study would be useful.
- The related work section of the paper needs to be extended, including other approaches that consider higher-order neighborhood information in the GNN.

[1] Christopher Morris, Martin Ritzert, Matthias Fey, William L. Hamilton, Jan Eric Lenssen, Gaurav Rattan, Martin Grohe. Weisfeiler and Leman Go Neural: Higher-order Graph Neural Networks. In AAAI, 2019.

[2] Johannes Klicpera, Aleksandar Bojchevski, and Stephan Gunnemann. Predict then propagate: Graph neural networks meet personalized pagerank. In ICLR, 2018.

[3] Guangtao Wang, Rex Ying, Jing Huang, Jure Leskovec. Multi-hop Attention Graph Neural Network. In IJCAI, 2021.

[4] Muhammet Balcilar, Guillaume Renton, Pierre Héroux, Benoit Gaüzère, Sébastien Adam, Paul Honeine. Analyzing the Expressive Power of Graph Neural Networks in a Spectral Perspective. In ICLR 2021.

---

### Decision · Action_Editors · 2023-06-07

**Recommendation:** Reject

**Comment:**

As mentioned above, all reviewers agree that the paper cannot be published in its current form in TMLR. One of the main issue is lack of mention and comparison with previous methods exploring the problem of incorporating high-order information in GNNs. As mentioned by reviewer P3uD, _the claimed contribution itself has been studied for a few years and it is quite a mature subject_, and the paper does not position itself clearly with respect to the literature on the topic.

**Audience:**

Potentially, the topic is relevant to the TMLR community.


**Claims And Evidence:**

All reviewers raised the issue that the approach taken by the paper has already been relatively thoroughly explored in the literature, and the submission does mention and connect with this existing literature.